# Rigorous Runtime Analysis of MOEA/D for Solving Multi-Objective Minimum Weight Base Problems

**Anh Viet Do**
Optimisation and Logistics
The University of Adelaide, Australia

**Aneta Neumann**
Optimisation and Logistics
The University of Adelaide, Australia

**Frank Neumann**
Optimisation and Logistics
The University of Adelaide, Australia

**Andrew M. Sutton**
Department of Computer Science
University of Minnesota Duluth, USA

## Abstract

We study the multi-objective minimum weight base problem, an abstraction of classical NP-hard combinatorial problems such as the multi-objective minimum spanning tree problem. We prove some important properties of the convex hull of the non-dominated front, such as its approximation quality and an upper bound on the number of extreme points. Using these properties, we give the first run-time analysis of the MOEA/D algorithm for this problem, an evolutionary algorithm that effectively optimizes by decomposing the objectives into single-objective components. We show that the MOEA/D, given an appropriate decomposition setting, finds all extreme points within expected fixed-parameter polynomial time, in the oracle model. Experiments are conducted on random bi-objective minimum spanning tree instances, and the results agree with our theoretical findings. Furthermore, compared with a previously studied evolutionary algorithm for the problem GSEMO, MOEA/D finds all extreme points much faster across all instances.

## 1 Introduction

Evolutionary algorithms have been widely used to tackle multi-objective optimization problems in many areas such as robotics, pattern recognition, data mining, bioinformatics, scheduling and planning, and neural network training [35]. Their population-based search operators make them a natural choice for simultaneously handling several possibly conflicting objectives. Many generic evolutionary multi-objective frameworks have been developed to supply basic implementations for any problem, and to provide templates that can be fine-tuned for specific applications (we refer to [30] for an overview of common approaches). Such features, along with their strong empirical performances in challenging applications, have led them to becoming one of the most attractive topics to researchers and practitioners alike.

Among evolutionary multi-objective algorithms (EMOs), arguably the most exemplary are dominance-based approaches such as GSEMO and NSGA variants, with the former often being considered a baseline. Another popular technique for multi-objective optimization is to decompose the multiple objectives into a single-objective subproblem. The MOEA/D algorithm is a state-of-the-art application of this technique in evolutionary computation [29, 31].

Despite the prevalence of EMOs on practical applications, rigorous analyses of their runtime behavior on meaningful problems are scarce. Nevertheless, these kinds of analyses are critical for (1) providing performance guarantees and guidelines to practitioners who use and develop these techniques in the field, and (2) promoting the explainability of heuristic search and optimization techniques by

37th Conference on Neural Information Processing Systems (NeurIPS 2023).

clarifying their working principles through a careful mathematical analysis. Run-time analyses on the performance of evolutionary algorithms have been provided for simple algorithms such as GSEMO in both artificial benchmark problems [2, 7] and others such as bi-objective minimum spanning tree [18, 28] and constrained submodular optimization [21, 22, 5, 23]. In recent years, theoretical analyses of state of the art approaches such as NSGA-II and MOEA/D have been conducted [12, 14, 13, 34, 6]. However, these run-time results have only been given for artificial benchmark problems.

In this paper, we present for the first time rigorous results on MOEA/D for a classical multi-objective optimization problem, namely the multi-objective minimum weight base problem. This problem, falling under the matroid optimization category, significantly generalizes the previously studied bi-objective minimum spanning tree problem. In this work, we focus on approximating the non-dominated front, as its size can be exponential in the problem size. In particular, we show that MOEA/D obtains a factor 2-approximation for two objectives in expected polynomial time. Previous analyses for the special case of graphic matroid (i.e. spanning forests) were only able to show a pseudo-polynomial run-time for GSEMO and NSGA-II to obtain this approximation [18, 4]. We further extend the analyses by deriving a fixed-parameter polynomial expected run-time in instances with $k \geq 2$ objectives to reach a $k$-approximation. That is, unlike the previous runtime bounds, ours is both polynomial and weight-free under fixed $k$ in light of the findings by Rechel et. al. [27].

Instrumental to our analyses is a deeper understanding of the problem, and as such, we formally examine certain properties of the multi-objective minimum weight base problem. We first prove a tight approximation guarantee from computing the convex hull of the non-dominated front, extending the known guarantee for two objectives [18]. With this in mind, we explore insight regarding this convex hull, including its vertex complexity and the structural relation among solutions whose weights constitute said convex hull. In addition, we briefly formulate an efficient deterministic approach to enumerate extreme points, which achieves a smaller approximation factor in lower (and weight-free) runtime than a recently proposed framework for general multi-objective minimization instances [1]. These findings may be of interest in areas beyond runtime analysis.

## 2 Preliminaries & Problem

First, we give an overview of relevant matroid theory concepts, with terminologies adopted from the well-known text book [20] on the subject.

**Definition 1.** *A tuple $M = (E, \mathcal{I} \subseteq 2^E)$ is a* matroid *if a)* $\emptyset \in \mathcal{I}$, *b)* $\forall x \subseteq y \subseteq E, y \in \mathcal{I} \implies x \in \mathcal{I}$, *c)* $\forall x, y \in \mathcal{I}, |x| < |y| \implies \exists e \in y \setminus x, x \cup \{e\} \in \mathcal{I}$. *The set $E$ is the ground set, and $\mathcal{I}$ is the independence collection. A* base *of $M$ is a maximal set in $\mathcal{I}$.*

**Definition 2.** *Given a matroid $M = (E, \mathcal{I})$, its* rank function, $r : 2^E \to \mathbb{N}$, *is defined as $r(x) = \max\{|y| : y \in 2^x \cap \mathcal{I}\}$, and the rank of $M$ is $r(E)$. A matroid is completely characterized by its rank function.*

To give examples, a $K$-rank uniform matroid over $E$ admits the independence collection $\mathcal{I} = \{x \subseteq E : |x| \leq K\}$, characterizing a cardinality constraint. In linear algebra, a representable matroid describes linear independence among a vector set. In graph theory, given an undirected graph $G = (V, E)$, a graphic matroid $M = (E, \mathcal{I})$ defined by $G$ is such that $\mathcal{I}$ contains all edge sets $x$ forming a forest subgraph in $G$. A base of a graphic matroid is a spanning forest, which itself is an object of much interest. Dual to the graphic matroid, the bond matroid $M^* = (E, \mathcal{I}^*)$ is such that $\mathcal{I}^*$ contains all edge sets $x$ whose removal from $E$ preserves every pairwise connectivity in $G$. The matroid properties emerge in many combinatorial structures of various optimization problems [20].

A classical application of matroids in optimization is in the minimum weight base (MWB) problem. Given a weighted matroid $(E, r, w)$, this problem asks to find a base in this matroid minimizing $w$. The most arguably well-known special case of MWB problem is the minimum spanning tree (MST) problem. It is known that the classical Greedy algorithm minimizes (and maximizes) arbitrary weight over a base collection of any matroid [24, 9, 8]. From the exchange property between independent sets (specifically the symmetric-exchange property proven in [3]), we see that Greedy can also enumerate all minimum weight bases, thus characterizes the optimality of any MWB instance. A proof of this quality is also included in [8].

The multi-objective minimum weight base (MOMWB) is a natural multi-objective extension to MWB. Given a $k$-weighted[1] matroid $(E, r, w \in (\mathbb{N}^*)^{k \times |E|})$ where $E$ is the ground set, $r$ is the rank function of the matroid, and the weight vector of a solution $x \in \{0, 1\}^{|E|}$ is $wx$ (also called the image of $x$ under $w$), the multi-objective problem asks to find a non-dominated set of bases in $(E, r)$ minimizing $w$. Here, $k$ is the number of objectives.

We denote $m := |E|$, $n := r(E)$, and observe that $x$ is a base in $(E, r)$ implies $|x| = n$. Given an objective vector function $f$ and solutions $x$ and $y$, $x$ dominates $y$, denoted with $x \preceq_f y$, iff $f(y) - f(x) \in \mathbb{R}^k_{\geq 0}$. We see that $x \preceq_f y$ iff $\min_{\lambda \in [0,1]^k} \lambda^\top (f(y) - f(x)) \geq 0$, i.e. $y$ has greater scalarized objective value than $x$ across all linear trade-offs. We denote the set of images of non-dominated solutions with $F$, and vertices of its convex hull $\mathrm{Conv}(F)$ are called *extreme points*. For convenience, let $\mathrm{Conv}(F)$ contain only points in $F$ and that its faces be conventionally defined, i.e. as continuous Euclidean subspaces.

Since $F$ can be exponentially large, we consider instead approximating it by finding solutions mapped to $\mathrm{Conv}(F)$. Such a set is known to guarantee a 2-approximation of the non-dominated set for $k = 2$ [18] under the following definition.

**Definition 3** (Minimization). *Given $k$ non-negative objective functions $f := (f_i)_{i=1}^k$, a solution $x$ $c$-approximates a solution $y$ for some $c \geq 0$ if $c f(y) - f(x) \in \mathbb{R}^k_{\geq 0}$. A solution set $X$ $c$-approximates (or is a $c$-approximation of) a solution set $Y$ if every $y \in Y$ is $c$-approximated by at least a $x \in X$.*

We formally describe categories of solutions of interest. Here, we only consider feasible solutions, e.g. bases in a MWB or MOMWB instance. Furthermore, a subset of $E$ is characterized by a bit-string in $\{0, 1\}^{|E|}$, so both set and bit operations on solutions are well-defined, and we use both representations throughout the paper.

**Definition 4.** *A solution $x$ is a supported solution to an instance with objective functions $f = (f_i)_{i=1}^k$ and a solution set $S$ if there is a linear trade-off $\lambda \in [0, 1]^k \setminus \{0\}$ where $x \in \mathrm{argmin}_{y \in S} \lambda^\top f(y)$. A trade-off set $\Lambda$ is complete if $\bigcup_{\lambda \in \Lambda} \mathrm{argmin}_{y \in S} \lambda^\top f(y)$ contains all supported solutions. A supported solution $z$ is extreme if there is $\lambda' \in [0, 1]^k$ where for all $x \in \mathrm{argmin}_{y \in S} \lambda'^\top f(y)$, $f(z) = f(x)$. A set containing a trade-off for each extreme solution is called sufficient.*

Note we assume that every instance admits a solution minimizing $\lambda^\top f$ for every $\lambda \in [0, 1]^k$. Of course, this holds for MOMWB due to the solution set being finite. We see that supported solutions are precisely the solutions whose images lie on $\mathrm{Conv}(F)$. Intuitively, a complete trade-off set decomposes the multi-objective instance in such a way to allow enumerating all supported solutions via exactly solving scalarized instances. Since supported solutions that are not extreme are mapped to points on the faces of $\mathrm{Conv}(F)$, we have the following observation.

**Observation 1.** *For each supported solution minimizing $\lambda^\top f$, there is an extreme solution minimizing $\lambda^\top f$. For every $\lambda \in [0, 1]^k$, there is an extreme solution minimizing $\lambda^\top f$.*

However, for linear functions, the number of supported solutions can be very large, so we also consider finding a representative subset which, as we will see, is sufficient to give an approximation guarantee.

**Definition 5.** *A solution set $X$ is sufficient to an instance with objective functions $f$ if for every extreme solution $y$, there is $x \in X$ where $f(x) = f(y)$. The analogy for supported solutions is called a complete solution set.*

With this definition, the set of solutions that are mapped to the extreme points is sufficient. In fact, the size of a minimal sufficient set is exactly the number of extreme points. Note that while the set of all supported solutions is unique, there can be multiple distinct minimal sufficient sets due to duplicate images. We briefly prove an approximation factor by any sufficient set, which is not restricted to MOMWB.

**Theorem 1** (Minimization). *Given $k \geq 1$ and a non-negative $k$-objective instance where for each objective $i$ there is $\delta_i > 0$ so that $f_i(x) \notin (0, \delta_i]$ for all solutions $x$, every sufficient solution set $P$ $k$-approximates all solutions. This factor is tight for all $k$, even if $P$ is a complete solution set.*

---

[1]The integrality does not affect the algorithms' behaviors, and is used only to ease the analysis. The positivity assumption ensures that approximation factors are meaningful.

We denote the weight scalarization with trade-off $\lambda$ with $w^{(\lambda)} := \lambda^\intercal w$, so $(E, r, w^{(\lambda)})$ is a scalarized instance at $\lambda$. All proofs, including the one for the above result, are included in the Appendix.

# 3 Properties of Conv($F$) in Multi-Objective Minimum Weight Base Problem

Here, we derive various properties of $\mathrm{Conv}(F)$ with implications on the complexity of approximating $F$. Since these solutions are optima of linearly scalarized instances, we use the properties of the Greedy algorithm, known to guarantee and characterize optimality in linear optimization over a matroid.

The Greedy algorithm starts from an empty set and adds elements to it in increasing weight order while maintaining its independence, until a base is reached. In essence, Greedy operates on a permutation over $E$ and produces a unique solution so we can characterize its outputs via permutations. We say a permutation $\tau$ over $E$, $\tau : E \rightarrow \{1, \ldots, m\}$, *sorts* the weight $w$ if, for all $i = 1, \ldots, m - 1$, $w_{\tau^{-1}(i)} \leq w_{\tau^{-1}(i+1)}$. As mentioned, Greedy run on a permutation that sorts the weight to be minimized returns a minimum weight base. More importantly, all minimum weight bases can be obtained by running Greedy on all sorting permutations. This allows us to derive properties of any solution mapped onto $\mathrm{Conv}(F)$ using Greedy's behaviors. In particular, we can circumvent the difficulty of examining weights by examining permutations instead, essentially looking at the weight-induced rankings rather than the weights themselves. As such, all results in this section are weight-free and hold for arbitrary real weights. We refer to [1] for a weight-dependent algorithmic treatment of $\mathrm{Conv}(F)$ under general settings.

**Observation 2.** *A MOMWB instance defined over a ground set $E$ satisfies the following: (1) its objective functions are linear, and (2) there is a surjective mapping from the set of permutations over $E$ to the set of supported solutions.*

To simplify analysis, we restrict the trade-off space to non-negative 1-norm unit vectors $U = \left\{a \in [0,1]^k : \sum_{i=1}^k a_i = 1\right\}$, let $\pi_\lambda$ be a permutation sorting $w^{(\lambda)}$ for $\lambda \in U$. For each $i \in E$, let $\mathbb{w}_i = (w_{j,i})_{j=1}^k$, and for each pair $i, j \in E$, let $\delta_{i,j} = \mathbb{w}_i - \mathbb{w}_j$ and $\Delta_{i,j} = \left\{a \in U : \delta_{i,j}^\intercal a = 0\right\}$ be the $(k-2)$-dimensional set characterized by the fact that for all $\lambda \in U$, $w_i^{(\lambda)} = w_j^{(\lambda)}$ iff $\lambda \in \Delta_{i,j}$. Finally, let $A$ be the multiset of non-empty $\Delta_{i,j}$ where $\delta_{i,j} \neq 0$, $H_A$ be the multiset of convex $(k-1)$-polytopes in $U$ defined by intersections of half-spaces bounded by hyperplanes in $A$ and the boundary of $U$, and $A'$ be the set of points in $U$ where each point lies in the interior of a polytope in $H_A$, we show that $A$ and $A'$ encompass complete solution set and sufficient solution set, respectively. Note that if $\delta_{i,j} = \mathbb{w}_i - \mathbb{w}_j = 0$, the inclusion of either $i$ or $j$ in a solution does not change its image under $w$.[2]

**Lemma 1.** *For any $Q \in H_A$, the set of all bases minimizing $w^{(\lambda)}$ remains constant for all $\lambda \in \mathrm{Int}(Q)$[3], and these bases share an image under $w$. Furthermore, they also minimize $w^{(\lambda)}$ for all $\lambda \in Q$.*

This immediately gives the upper bound on the number of extreme points, which is the maximum number of space partitions by hyperplanes; the formula for this is given in [32]. Note that Lemma 1 only requires properties in Observation 2, which hold for the broader class of set systems that is matroid embeddings since all minimum weight bases in such a system are exactly Greedy bases [11].

**Corollary 1.** *The size of a minimal sufficient solution set to a $k$-objective instance satisfying properties in Observation 2 is at most $\sum_{i=1}^k \binom{m(m-1)/2}{i-1}$, and $A'$ is a sufficient trade-off set.*

We remark that we deliberately choose each trade-off in $A'$ from the interior of each polytope. This is because if a zero trade-off coefficient is assigned to an objective, then bases minimizing the weight scalarized by such a trade-off may not be non-dominated. Furthermore, such scalarized weights admit optima whose images under $w$ are identical, which is necessary to ensure that the first optimum an optimization algorithm finds using these trade-offs is an extreme solution. Moreover, this trade-off

---

[2]We include polytopes with empty interiors in $H_A$ to account for overlapping hyperplanes in $A$. Furthermore, we assume these hyperplanes are ordered arbitrarily along their normal direction for the purpose of defining interior-free polytopes: $h$ such hyperplanes form $h - 1$ polytopes.

[3]Given a set $A$ in a metric space, $\mathrm{Int}(A)$ is the set of its interior points.

selection scheme also guarantees that said algorithm does not discard vertices of $\text{Conv}(F)$ over time, unless it stores all found optima.

Given a solution set $S$, the $l$-Hamming neighborhood graph of $S$ is an undirected graph $G_l = (S, \{\{a, b\} : |a \otimes b| \leq l\})$, and $S$ is $l$-Hamming connected if $G_l$ is connected. Neumann [18] proved for spanning trees that given the non-dominated front being strongly convex, the set of supported solutions is 2-Hamming connected. We show that this even holds for matroid bases without the convexity assumption. For simplicity, we assume, for the rest of the analysis, fixed orderings among each class of elements $i \in E$ sharing $\mathbb{w}_i$. We will see that the existence of such elements does not affect the 2-Hamming connectivity among supported solutions.

We first show that as the trade-off moves continuously within $U$, the permutation sorting the scalarized weight is transformed incrementally by exchanging two adjacent positions, which we call an adjacent swap.

**Lemma 2.** *For any $a, a' \in U$, let $A^* = \{a_i\}_{i=1}^h$ be the multiset of intersections between the line segment connecting $a$ and $a'$ and hyperplanes in $A$, indexed in the order from $a$ to $a'$, there is a shortest sequence of adjacent swaps from $\pi_a$ to $\pi_{a'}$, $(\pi_a, \tau_1, \ldots, \tau_h, \pi_{a'})$, where for all $i = 1, \ldots, h$, $\tau_i$ sorts $w^{(a_i)}$. If $w^{(a)}$ or $w^{(a')}$ can be sorted by multiple permutations, the claim holds assuming that $\pi_a$ and $\pi_{a'}$ have maximum Kendall distance[4].*

Next, we show that an adjacent swap on the sorting permutation incurs an at most 2-bit change in the minimum weight base.

**Lemma 3.** *Let $\tau$ and $\tau'$ be permutations over $E$ that are one adjacent swap apart, and $x$ and $x'$ are Greedy solutions on them, respectively, then $|x \otimes x'| \leq 2$. Furthermore, let $u, v \in E$ where $\tau(v) = \tau(u) + 1$ and $\tau'(v) = \tau'(u) - 1$, $|x \otimes x'| = 2$ iff $x' = x \setminus \{u\} \cup \{v\}$.*

Lemma 2 and 3 indicate that there is a sequence of 2-bit flips between any pair of supported solutions such that every step also gives a supported solution. Therefore, starting from a supported solution, we can compute the rest of $\text{Conv}(F)$ with 2-bit variations.

Regarding weight-sharing elements, for a supported solution $x$ minimizing $w^{(\lambda)}$, if there is a class of equal-weight elements $Z$ partially intersecting $x$, then all supported solutions minimizing $w^{(\lambda)}$ containing different elements in $Z$ can be reached from $x$ by a sequence of 2-bit flips, each step in which produces a supported solution also minimizing $w^{(\lambda)}$. This is because $Z$ is located consecutively in $\pi_\lambda$ and can be arranged arbitrarily (leading to the Greedy solution minimizing $w^{(\lambda)}$), and there is a sequence of adjacent swaps between any two such permutations touching only elements in $Z$. Furthermore, if there are multiple such classes whose elements share a scalarized weight at some trade-off $\lambda$, the relative inter-class orderings in any valid $\pi_\lambda$ can be shuffled arbitrarily with an adjacent swap sequence that neither, at any step, changes any pairwise intra-class ordering, nor breaks the sorting property. For these two reasons, the set of permutations sorting all scalarized weights is 1-Kendall connected (Kendall-distance equivalence to Hamming connectivity), thus the relative intra-class orderings, and consequentially the presence of multiple elements within each such class, does not affect 2-Hamming connectivity.

**Corollary 2.** *Given solutions $x$ and $y$ where $\{wx, wy\} \subseteq \text{Conv}(F)$, there is a non-empty set of solutions $\{z_i\}_{i=1}^h$ where $x = z_1$, $y = z_h$, $|z_i \otimes z_{i+1}| = 2$ for all $i = 1, \ldots, h-1$ and $\{wz_i\}_{i=1}^h \subseteq \text{Conv}(F)$.*

Lemma 3 also lets us derive a stronger bound on the number of distinct Greedy solutions as the trade-off moves in a straight line, giving an upper bound on the number of extreme points in case $k = 2$.

**Theorem 2.** *Given $n \in (0, m)$, $a, b \in U$ and $X$ is a minimal set of extreme solutions such that for each $\theta \in [0, 1]$, $X$ contains a solution minimizing $w^{((1-\theta)a+\theta b)}$, $|X| \leq hm - h(h+1)/2 + 1$ where $h := \left\lceil \sqrt{2 \min\{n, m-n\} - 1} \right\rceil$.*

**Corollary 3.** *A bi-objective MWB instance (i.e. $k = 2$) admits at most $O(m\sqrt{\min\{n, m-n\}})$ extreme points.*

We remark that aside from the trivial cases $n \in \{1, m-1\}$, we did not find an instance where this bound is tight. As far as we are aware, it is an open question whether this bound is optimal.

---

[4]Kendall distance between two permutations equals the minimum number of adjacent swaps needed to transform one into the other [17].

**Algorithm 1:** Finding extreme points and a complete trade-off set (adapted from [10])

**Input:** Multi-weighted matroid $(E, r, w)$
**Output:** $S, \Lambda$

1 $S, \Lambda' \leftarrow \emptyset$;
2 $\Lambda \leftarrow \{e_i\}_{i=1}^k$;
3 $P \leftarrow$ all permutations over $\{1, \ldots, k\}$;
4 **while** $\Lambda \setminus \Lambda' \neq \emptyset$ **do**
5      **for** $\lambda \in \Lambda \setminus \Lambda'$ **do**
6          $\forall p \in P, a_p \leftarrow$ base minimizing $w^{(\lambda)}$ prioritizing weights ranked by $p$;
7          $S \leftarrow S \cup \{a_p\}_{p \in P}$;
8      $\Lambda' \leftarrow \Lambda' \cup \Lambda$;
9      $\Lambda \leftarrow$ non-negative normal vectors to facets of $\text{Conv}(\{wx : x \in S\})$;

---

**Algorithm 2:** Special case of Algorithm 1 for $k = 2$

**Input:** Bi-weighted matroid $(E, r, w_1, w_2)$
**Output:** $S, \Lambda$

1 $S, \Lambda' \leftarrow \emptyset$;
2 $\Lambda \leftarrow \{0, 1\}$;
3 **while** $\Lambda \setminus \Lambda' \neq \emptyset$ **do**
4      **for** $\lambda \in \Lambda \setminus \Lambda'$ **do**
5          $a, b \leftarrow$ bases minimizing $(1 - \lambda)w_1 + \lambda w_2$ prioritizing $w_1$ and $w_2$, respectively;
6          $S \leftarrow S \cup \{a, b\}$;
7      $\Lambda' \leftarrow \Lambda' \cup \Lambda$;
8      $\Lambda \leftarrow \emptyset$;
9      $\pi \leftarrow$ element indexes of $S$ in increasing $w_1(\cdot)$ order;
10      **for** $i \in \{1, \ldots, |S| - 1\}$ **do**
11          $\delta_1, \delta_2 \leftarrow w_1 S_{\pi(i+1)} - w_1 S_{\pi(i)}, w_2 S_{\pi(i)} - w_2 S_{\pi(i+1)}$;
12          $\Lambda \leftarrow \Lambda \cup \{\delta_1 / (\delta_1 + \delta_2)\}$;

## 4 Exact Computation of Extreme Points

In this section, we describe a deterministic framework that finds a solution for each extreme point, as well as a complete trade-off set. This framework, modified from the algorithm proposed in [10] for bi-objective MST, is outlined in Algorithm 1. It calls another algorithm (e.g. Greedy) to find MWB to scalarized weights, and iteratively computes new extreme points based on information from previous ones. Intuitively, each subset of extreme points $Z$ is such that its convex hull is "inside" $\text{Conv}(F)$ and contains, for each extreme point $y \notin Z$, a facet separating $Z$ from $y$. This means $y$ can be discovered by minimizing the weight scalarized along the normal direction of this facet, essentially expanding $\text{Conv}(Z)$ to "touch" $y$. This iterative process begins with an optimum in each objective, and ends when all new normal vectors are duplicates of ones found previously, indicating that the current convex hull cannot be expanded further and equals $\text{Conv}(F)$. The special case of this algorithm for $k = 2$ is given in Algorithm 2 which treats trade-offs as scalars.

Algorithm 1, naively implemented, requires $O(\#(\text{poly}(k) + k!m \log m))$ operations and $O(k!\#m)$ calls to the matroid rank oracle where # is the number of extreme points. Each iteration in the main loop adds at least one extreme point, and redundant points are excluded from future iterations via $\Lambda'$. Here, we assume updating trade-off for each new vertex takes $\text{poly}(k)$ operations.

We remark that exhaustive tie-breaking over all objectives is done at line 6 to ensure that the computed points are the vertices of $\text{Conv}(F)$ instead of interior points of its faces, and that all extreme points are accounted for when the termination criterion is met. Furthermore, if the trade-off assigns zero value to some objectives, this also guarantees that the resulted solutions are non-dominated. This subroutine can be improved by grouping together objectives whose sorting permutations agree (in relative orderings) among elements tied by $w^{(\lambda)}$.

---
**Algorithm 3:** MOEA/D for MOMWB
---
**Input:** A MOMWB instance, trade-off set $\Lambda$, neighborhood size $N \geq 1$
**Output:** $S$
1  $\forall \lambda \in \Lambda, B_\lambda \leftarrow N$ nearest neighbors of $\lambda$ in $\Lambda$ (Euclidean distance);
2  $\forall \lambda \in \Lambda, P_\lambda \leftarrow$ a random sample from $\{0,1\}^m$;
3  $S \leftarrow \emptyset$;
4  **while** *stopping conditions not met* **do**
5  $\quad$ **for** $\lambda \in \Lambda$ **do**
6  $\quad\quad$ $x \leftarrow$ uniformly sampled from $P_\lambda$;
7  $\quad\quad$ $y \leftarrow$ independent bit flips on $x$ with probability $1/m$;
8  $\quad\quad$ $D \leftarrow \{l \in B_\lambda : \forall z \in P_l, f_l(y) < f_l(z)\}$;
9  $\quad\quad$ $T \leftarrow \{l \in B_\lambda : \forall z \in P_l, f_l(y) = f_l(z)\}$;
10 $\quad\quad$ $\forall l \in D, P_l \leftarrow \{y\}$;
11 $\quad\quad$ $\forall l \in T, P_l \leftarrow P_l \cup \{y\}$ removing solutions with duplicate images;
12 $\quad\quad$ $S \leftarrow$ non-dominated individuals in $S \cup \{y\}$ under relation $\preceq_g$;
---

## 5   MOEA/D With Weight Scalarization

Multi-Objective Evolutionary Algorithm based on Decomposition (MOEA/D), introduced in [33], is a co-evolutionary framework characterized by simultaneous optimization of single-objective subproblems in a multi-objective problem. While there are many approaches to decompose the multi-objective into single-objectives, we consider the classical approach that is weight scalarization [15], as hinted in preceding sections. This simple scheme is sufficient in approximating $F$ and even enumerating $\text{Conv}(F)$.

### 5.1   Description

MOEA/D uses two fitness functions, a scalar function formulated by the decomposition scheme and a vector function for dominance checking [33]. To account for the matroid base constraint, we use the penalty term formulated in [26], which was adapted from prior work on MST [19]. Letting $w_{max} := \max_{(i,e) \in \{1,...,k\} \times E} w_{i,e}$, we have the fitness $f_\lambda$ of $x \in \{0,1\}^m$ at trade-off $\lambda$, and the fitness vector $g$ for dominance checking where $\mathbb{1}$ is the one vector.

$$f_\lambda(x) := m(n - r(x))w_{max} + w^{(\lambda)}x, \quad g(x) := m(n - r(x))w_{max}\mathbb{1} + wx \qquad (1)$$

The MOEA/D for the MOMWB problem is outlined in Algorithm 3. The fitness functions defined in Eq. (1) and the input trade-off set realize the decomposition, and the algorithm evolves a population for each scalarized subproblem with potential neighborhood-based collaboration. During the search, it maintains a non-dominated solution set $S$, which does not influence the search and is returned as output. An optimum to each scalarized subproblem is a supported solution. Note in this formulation, MOEA/D keeps ties in each subproblem, allowing all found optima to participate in mutation operations. This is to avoid having to flip more than two bits to jump from a supported solution to an uncollected point in $\text{Conv}(F)$. We will see that while this may increase the population size, it does not affect the run-time to reach optimality in each subproblem.

### 5.2   Expected Time To Minimize Scalarized Weights

In the following, we do not assume a particular value of $N$, and the results hold for any $N \geq 1$. Furthermore, we exclude the trivial instances with $n = m$ and $n = 0$, which admit exactly one base each.

**Lemma 4.** *MOEA/D working on trade-off set $\Lambda$ finds a base's superset for each $\lambda \in \Lambda$ within $O(|\Lambda|m \log n)$ expected search points.*

Let $OPT_\lambda$ be the optimal value to $(E, r, w^{(\lambda)})$, we have the following drift argument proven in [26] for the standard bit mutation in the MWB problem.

**Lemma 5** ([26], Proposition 9). *Given a trade-off $\lambda \in [0,1]$ and $x \in \{0,1\}^m$, if $x$ supersets a base, then there are $n$ 2-bit flips and $m - n$ 1-bit flips on $x$ reducing $f_\lambda(x)$ on average by $(f_\lambda(x) - OPT_\lambda)/m$.*

We use the same ideas as the proof of Theorem 2 in [26], while sharpening an argument to derive a slightly tighter bound.

**Theorem 3.** *MOEA/D working on trade-off set $\Lambda$ finds MWBs to instances scalarized by trade-offs in $\Lambda$ in $O\left(|\Lambda|m\log n + \sum_{\lambda \in \Lambda} m^2(\log(m-n) + \log w_{max} + \log d_\lambda)\right)$ expected search points where $d_\lambda := \min\{a > 0 : a\lambda, a(1-\lambda) \in \mathbb{N}\}$.*

In order for MOEA/D to reach a $k$-approximation and not lose it afterwards, it suffices that $\Lambda$ is sufficient and each scalarized subproblem admits optima with a unique image. As mentioned and from Lemma 1, this can be obtained by sampling from the interiors of convex polytopes in $H_A$. For $k = 2$, this can be done by taking a complete scalar trade-off set $A$ (e.g. as returned by Algorithm 2) and include $(a+b)/2$ (which is an interior point) for each non-empty interval $(a,b)$ bounded by consecutive elements in $A \cup \{0,1\}$. Under the integer weights assumption, this method gives rational trade-offs whose integral denominators are $O(w_{max}^2)$, so we have the following bound from Corollary 3.

**Corollary 4.** *For a bi-objective instance, MOEA/D working on a minimal sufficient trade-off set finds a sufficient solution set within $O(m^2\sqrt{\min\{n, m-n\}}(m(\log(m-n) + 3\log w_{max}) + \log n))$ expected number of search points.*

This method can be generalized to higher dimensions. Instead of taking an average of two consecutive elements, we can take the average of the vectors normal to $k$ facets of $\mathrm{Conv}(F)$ that meet at an extreme point, as this ensures the resulted trade-off $\lambda$ is not normal[5] to any $h$-faces of $\mathrm{Conv}(F)$ for all $h \in [1, k]$, and that said extreme point minimizes $w^{(\lambda)}$. Since each facet is determinable by $k$ points with integral coordinates, each such (1-norm unit) vector admits rational coordinates with denominator at most $kw_{max}$. Therefore, the trade-offs derived by this method admit rational representations whose denominators are $O(k^{k+1}w_{max}^k)$, giving the run-time upper bound from Corollary 1 under the assumption that $k$ is sufficiently small.

**Corollary 5.** *Given a $k$-obbjective instance where $k \in o(m)$ and $k \in o(w_{max})$, MOEA/D working on a minimal sufficient trade-off set guarantees $k$-approximation within $O(m^{2k-1}(m(\log(m-n) + (k+1)\log w_{max}) + \log n))$ expected number of search points.*

As a side note, since MOEA/D uses standard bit mutation, we can replace $w_{max}$ with $m^m$ and remove $\log d_\lambda$ from the bound in Theorem 3 to arrive at weight-free asymptotic bounds [27].

## 5.3 Expected Time To Enumerate Conv($F$)

We see from Corollary 2 that MOEA/D with a complete trade-off set can collect all points in $\mathrm{Conv}(F)$ with 2-bit flips starting from an optimum to each subproblem. As mentioned, this is afforded by allowing all found optima to undergo mutation.

**Theorem 4.** *Assuming distinct supported solutions have distinct images under $w$, MOEA/D working on a minimal complete trade-off set $\Lambda$, and starting from an optimum for each $\lambda \in \Lambda$ enumerates $C := \mathrm{Conv}(F)$ in $O(|\Lambda||C|^2m^2)$ expected number of search points.*

With this, Theorem 3 and Corollary 3 and 1, we have the following expected run-time bounds under the distinct image assumption. Note this assumption can be removed by having MOEA/D keep duplicate images at line 11.

**Corollary 6.** *For a bi-objective instance, MOEA/D working on a minimal complete trade-off set enumerates $C := \mathrm{Conv}(F)$ in expected time $O(m^2\sqrt{\min\{n, m-n\}}(m(\log(m-n) + 3\log w_{max} + |C|^2) + \log n))$.*

**Corollary 7.** *Given a $k$-objective instance where $k \in o(m)$ and $k \in o(w_{max})$, MOEA/D working on a minimal complete trade-off set enumerates $C := \mathrm{Conv}(F)$ in expected time $O(m^{2k-1}(m(\log(m-n) + (k+1)\log w_{max} + |C|^2) + \log n))$.*

---

[5]If $\lambda$ is normal to a face of $\mathrm{Conv}(F)$, then all points in this face minimize $w^{(\lambda)}$. A convex combination of vectors normal to some faces is normal to their (assumed non-empty) intersection, the intersection of $h$ adjacent $d$-faces is a $(d-h+1)$-face, and an extreme point or vertex is a 0-face.

# 6 Experimental Investigation

In this section, we perform computational runs of MOEA/D on various bi-objective minimum spanning tree instances. Spanning trees in a connected graph $G = (V, E)$ are bases of the graphic matroid defined by $G$ admitting the ground set $E$. The rank of such a matroid (i.e. the size of the spanning tree) equals $|V| - 1$ and its rank function is defined with $r(x) = |V| - cc(x)$ where $cc(x)$ is the number of connected components in $(V, x)$. In notations, we have $n = |V| - 1$ and $m = |E|$. We use simple undirected graphs in our experiments, and the edge-set representation of solutions in the implementations of the algorithms [25].

## 6.1 Setting and Performance Metrics

We uniformly sample graphs with $|V| \in \{26, 51, 101\}$ and $|E| \in \{150, 300\}$. In this procedure, edges are added randomly into an empty graph up to the desired edge count, and this is repeated until a connected graph is obtained. Each edge weight is an integer sampled independently from $\mathcal{U}(1, 100)$. We generate two weighted graphs with each setting, making 12 instances in total.

With this experiment, we aim to measure the algorithms' performances in finding solutions mapped to all extreme points, we denote this set of target points with $R$. We compare MOEA/D against GSEMO, previously studied for its performance in bi-objective MST [18]. For GSEMO, we use the fitness function $g$ defined in Eq. (1). Since GSEMO checks for dominance in each iteration across its entire population, we set $N := |\Lambda|$ for MOEA/D to match. Here, the input $\Lambda$ to MOEA/D is derived from a complete trade-off set output by Algorithm 2 in the manner described in Section 5.2. The points given by Algorithm 1 are the target points $R$, and each run is terminated if all target points are hit. Additionally, each run is terminated after at most $\lceil 3|R|m^2 \log(m - n) \rceil$ evaluations (each evaluation is a call to function $g$). Each algorithm is run on each instance 10 times. Their performances are measured with the followings ($S$ is the final population returned by the algorithm):

- **Success rate:** The number of runs where all target points are hit within the evaluation budget.

- **Cover rate:** The proportion of hit target points after termination, $|R \cap \{wx : x \in S\}|/|R|$. A run is successful if this reaches 100%.

- **Modified inverted generational distance (IGD+):** The distance between the output and the target points [16], $\sum_{y \in R} \min_{x \in S} \sqrt{\sum_{i=1}^{k} (\max\{(wx)_i - y_i, 0\})^2}/|R|$. A run is successful if this reaches 0.

- **T:** The number of evaluations until all target points are hit.

We remark that the fitnesses $f_\lambda$ can be quickly computed alongside $g$, incurring minimal overheads. In fact, the run-time bottleneck is in checking the number of connected components.

## 6.2 Experimental Results

The results are shown in Table 1, with IGD+ and cover rate from MOEA/D omitted due to them being 0 and 100% across all instances, respectively. These are contextualized by the listed instance-specific parameters. Of note is the number of target points $|R|$ which is smaller than the upper bound in Theorem 2 in all instances.

Immediately, we see the GSEMO failed to hit all target points within the evaluation budget in most runs, while MOEA/D succeeded in every run. In most cases, GSEMO hit at most one target point. Inspecting the output points and IGD+ reveals that its population converged well to the non-dominated front, yet somehow misses most extreme points. In contrast, MOEA/D hit all target points within up to 92% of the evaluation budget, though there are significant relative variances in the run-times.

Inspecting the run-times of MOEA/D in relation to the evaluation budgets, we see that the means of ratios remain fairly stable across instances. This suggests the asymptotic bound in Theorem 3 is not overly pessimistic. Instances 5 and 6 are particularly interesting as they are ostensibly the easiest due to the small number of extreme points, yet MOEA/D seems to require the most fractions of the budget. Given that these instances exhibit the smallest $m - n$, this can be explained by the interference of lower-order terms in the asymptotic bound, which are not counted in the budgets.

Table 1: Means and standard deviations of performance statistics from GSEMO and MOEA/D on bi-objective MST instances. Means and standard deviations of $T$ are computed over successful runs only. All differences are statistically significant.

| Id | $m$ | $n$ | $|R|$ | max eval. | GSEMO | | | | MOEA/D | |
|----|-----|-----|-------|-----------|-----------|------------|------------|--------------|-----------|-------------|
| | | | | | Suc. rate | Cover rate | IGD+ | T/max eval. | Suc. rate | T/max eval. |
| 1 | 150 | 25 | 39 | 12710536 | 2/10 | 2.5±0.06% | 0.13±0.11 | 79±11% | 10/10 | 50±15% |
| 2 | 150 | 25 | 31 | 10103247 | 0/10 | 3.1±0.14% | 0.25±0.32 | N/A | 10/10 | 43±11% |
| 3 | 150 | 50 | 45 | 13988205 | 0/10 | 1.9±0.1% | 1.7±0.7 | N/A | 10/10 | 52±11% |
| 4 | 150 | 50 | 45 | 13988205 | 0/10 | 2±0.083% | 0.92±0.23 | N/A | 10/10 | 47±12% |
| 5 | 150 | 100 | 35 | 9242155 | 0/10 | 2.5±0.13% | 2.6±1.2 | N/A | 10/10 | 63±15% |
| 6 | 150 | 100 | 36 | 9506216 | 0/10 | 2.2±0.21% | 3.7±2.3 | N/A | 10/10 | 58±11% |
| 7 | 300 | 25 | 45 | 68243769 | 3/10 | 2.2±0.031% | 0.06±0.072 | 86±13% | 10/10 | 41±11% |
| 8 | 300 | 25 | 49 | 74309882 | 1/10 | 2±0.031% | 0.076±0.083 | 94±0% | 10/10 | 40±8.1% |
| 9 | 300 | 50 | 66 | 98392434 | 0/10 | 1.4±0.048% | 0.49±0.13 | N/A | 10/10 | 50±11% |
| 10 | 300 | 50 | 63 | 93920051 | 0/10 | 1.5±0.045% | 0.8±0.24 | N/A | 10/10 | 60±15% |
| 11 | 300 | 100 | 79 | 113013110 | 0/10 | 1.1±0.043% | 2±0.53 | N/A | 10/10 | 60±8.3% |
| 12 | 300 | 100 | 80 | 114443656 | 0/10 | 1.1±0.072% | 3.1±1 | N/A | 10/10 | 58±11% |

## 7   Conclusion

In this study, we contribute to the theoretical analyses of evolutionary multi-objective optimization in the context of non-trivial combinatorial problems. We give the first run-time analysis of the MOEA/D algorithm for a broad problem class that is multi-objective minimum weight base problem. In particular, we show a fixed-parameter polynomial expected run-time for approximating the non-dominated front, simultaneously extending existing pseudo-polynomial bounds for GSEMO to arbitrary number of objectives and broader combinatorial structures. Our experiments in random bi-objective minimum spanning tree instances indicate that MOEA/D significantly outperforms GSEMO in the computing extreme points under an appropriate decomposition. Along the way, we prove properties that give further insight into the problem of interest.

## Acknowledgements

This work was supported by Australian Research Council grants DP190103894 and FT200100536, and by National Science Foundation grant 2144080.

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

## A Omitted proofs

*Proof of Theorem 1.* Let $f := (f_i)_{i=1}^k$ be the objective function vector, $z$ be any solution, $Z = \{i : f_i(z) = 0\}$, if $|Z| = k$ then $z$ is an extreme solution, so $P$ 1-approximates $z$. Assume otherwise, we define $\lambda \in (0,1]^k$ where $\lambda_i := \epsilon/\delta_i$ if $i \in Z$ and $\lambda_i := \epsilon/[(k-|Z|)f_i(z)]$ otherwise for some sufficiently small $\epsilon > 0$. By definition of sufficient solution set and Observation 1, there is $x \in P$ minimizing $\lambda^\top f$, i.e. $\lambda^\top f(x) \le \lambda^\top f(z) = \epsilon$. If $f_i(x) > 0$ for some $i \in Z$ or $f_i(x) > (k-|Z|)f_i(z)$ for some $i \notin Z$, then since $f(x) \in \mathbb{R}_{\ge 0}^k$, we have $\lambda^\top f(x) > \epsilon$, a contradiction. Therefore, $x$, and by extension $P$, $(k-|Z|)$-approximates $z$. Since $z$ can assume positive values in all objectives[6], this factor simplifies to $k$.

We show tightness by construction. Let $\epsilon \in (0,k)$, $m := k^2$, $\theta_i := \sum_{j=0}^{k-1} e_{ik-j}$ for $i = 1,\ldots,k$ where $e_j$ is the $j$th unit vector in $\mathbb{R}^m$, we define a non-negative $k$-objective instance over $\{0,1\}^m$: $\min_x\{f(x) := (\theta_i^\top x - \epsilon \prod_{j=0}^{k-1} x_{ik-j})_{i=1}^k : |x| \ge k\}$. We see that the set of all supported solutions is precisely $S := \{\theta_i\}_{i=1}^k$. Let $z := \sum_{i=0}^{k-1} e_{ik+1}$ be a solution, for all $i = 1,\ldots,k$, $f_i(\theta_i) = k - \epsilon \ge (k-\epsilon)f_i(z)$ (equality holds if $k > 1$). This means $S$ fails to $(k-\epsilon-\varepsilon)$-approximate $z$ for any $\varepsilon > 0$, and $\epsilon$ can be arbitrarily small. Since $S$ is a complete solution set, the claim follows. □

*Proof of Lemma 1.* Let $c$ be any point in $\mathrm{Int}(Q)$, by definition of $A$, $w_i^{(c)} = w_j^{(c)}$ iff $\delta_{i,j} = 0$, and $w^{(c)}$ admits multiple minima iff they contain different elements among those sharing weights in $w^{(c)}$, while sharing all other elements. Indeed, let $x$ and $y$ be a pair of minima violating this condition, they must contain different sets of weights so for all bijection $\gamma$ between $x \setminus y$ and $y \setminus x$, there is $u \in x \setminus y$ where $w_u^{(c)} \ne w_{\gamma(u)}^{(c)}$; this leads to a contradiction when combined with the base exchange property. This means these optima share image under $w$, and bases not having the same image do not minimize $w^{(c)}$.

Let $b$ be any point on the boundary of $Q$ and $L$ be the set of points between $b$ and $c$ excluding endpoints, we show that $\pi_c$ also sorts $w^{(b)}$. Let $i.j \in E$ where $w_i^{(c)} < w_j^{(c)}$, then $w_i^{(b)} > w_j^{(b)}$ implies $w_i^{(d)} = w_j^{(d)}$ for some $d \in L$, meaning $L$ meets a hyperplane in $A$, a contradiction as $L \subseteq \mathrm{Int}(Q)$. For all pairs $i, j \in E$ where $w_i^{(c)} = w_j^{(c)}$, $\delta_{i,j} = 0$ so $w_i^{(b)} = w_j^{(b)}$. With this, every pair is accounted for, so $\pi_c$ sorts $w^{(b)}$. Therefore, since Greedy guarantees optimality, any base minimizing $w^{(c)}$ also minimizes $w^{(b)}$, yielding the claim. □

*Proof of Corollary 1.* We see that $|A'| = |H_A|$, which is upper bounded by the number of half-space intersections from hyperplanes in $A$. Since these are $(k-2)$-dimensional hyperplanes, applying the formula in [32] gives $|H_A| \le \sum_{i=1}^k \binom{|A|}{i-1}$ which is increasing in $|A|$, so the claim follows from $|A| \le m(m-1)/2$. We have $A'$ is a sufficient trade-off set following from Lemma 1 and $\bigcup_{Q \in H_A} Q = U$. □

*Proof of Lemma 2.* Let $0 < \lambda_c, < \lambda_d < 1$ such that $b := (1-b)a + ba' \in A^*$ and for all $\lambda \in [\lambda_c, \lambda_b)$, $(1-\lambda)a + \lambda a' \notin A^*$, and let $c := (1-\lambda_c)a + \lambda_c a'$, then elements sharing weight in $w^{(b)}$ must be mapped to consecutive positions in $\pi_c$. Indeed, let $p, q \in E$ $(\pi_c(p) < \pi_c(q))$ where $w_p^{(b)} = w_q^{(b)}$, if there is $o \in E$ where $\pi_c(o) \in (\pi_c(p), \pi_c(q))$ and $w_o^{(b)} \ne w_p^{(b)}$, then since the former implies $w_o^{(c)} \in (w_p^{(c)}, w_q^{(c)})$, we have $w_o^{(d)} = w_p^{(d)}$ or $w_o^{(d)} = w_q^{(d)}$ for some $d$ in the open line segment connecting $b$ and $c$ which implies $d \in A^*$, a contradiction. Each such consecutive sequence of $l$ positions contains $l(l-1)/2$ pairs. From here, we consider two cases:

- If such a sequence contains no pair $(i, j)$ where $\delta_{i,j} = 0$, then the aforementioned pairs correspond to $l(l-1)/2$ duplicates of $b$ in $A^*$. Furthermore, since the weights are transformed linearly w.r.t. trade-off, for all sufficiently small $\epsilon > 0$, these sequences are reversed between $\pi_c$ and $\pi_{b+\epsilon(b-c)}$, whereas positions not in these sequences are stationary. Reversing $l$ consecutive positions requires $l(l-1)/2$ adjacent swaps, so the Kendall distance between $\pi_c$ and $\pi_{b+\epsilon(b-c)}$ equals the multiplicity of $b$ in $A^*$.

---

[6]If $|Z| \ge k'$ for all solutions $z$, the instance is reducible to $(k-k')$-objective instances, and the guarantee factor is likewise tight.

- If such a sequence contains $h > 1$ elements with the same weight at all trade-off, then these must occupy consecutive positions in $\pi_c$. As we assumed, the relative ordering among these elements is fixed, so exactly $h(h-1)/2$ swaps are saved. Furthermore, any pair $(i, j)$ among these elements is such that $\Delta_{i,j} \notin A$, meaning these $h(h-1)/2$ pairs are already subtracted from $A^*$.

In any case, we can assign to each duplicate of $b$ in $A^*$ a permutation sorting $w^{(b)}$ so that these form a sequence of adjacent swap from $\pi_c$ to $\pi_{b+\epsilon(b-c)}$ including $\pi_{b+\epsilon(b-c)}$ and not $\pi_c$. This directly yields the claim if $a$ and $a^*$ are not in $A^*$.

Assume otherwise, then for all hyperplanes $\Delta_{i,j}$ containing $a$, $w_i^{(a)} = w_j^{(a)}$, so for every such pair $(i, j)$, we arrange $\pi_a$ so that their pairwise ordering in $\pi_a$ is the reverse of that in $\pi_{a'}$. We likewise give $a'$ the same treatment[7]. With this, the Kendall distance between $\pi_a$ and $\pi_{a'}$ is maximized and equal to $|A^*|$. $\qquad\square$

*Proof of Lemma 3.* Let $E_o := \{a \in E : \tau(a) < \tau(o)\}$ be the set of elements Greedy considers adding to $x$ before $o \in E$ when run on $\tau$, we have $x \cap E_u = x' \cap E_u$. If $v \in x$ or $v \notin x'$ or $u \notin x$ or $u \in x'$ then $x = x'$, as can be seen from how Greedy selects elements:

- If $v \in x$, then $v \in x'$ since Greedy observes $v$ before $u$ when run on $\tau'$. Whether Greedy adds $u$ to $x$ only depends on whether there is a circuit in $(x \cap E_u) \cup \{u\} = (x' \cap E_u) \cup \{u\}$, so it makes the same decision when run on $\tau'$. Afterwards, it proceeds identically on both permutations, leading to the same outcome, so $x = x'$. By symmetry, the same follows from $u \in x'$.

- If $u \notin x$, then there is a circuit in $(x \cap E_u) \cup \{u\} = (x' \cap E_u) \cup \{u\}$, so $u \notin x'$. By the same argument, Greedy makes the same decision regarding $v$ on both permutations, leading to $x = x'$. By symmetry, the same follows from $v \notin x'$.

Assume otherwise, it is a known property of bases [20] that $x \cup \{v\}$ contains a unique circuit $C$ and that $v \in C$. Greedy not adding $v$ to $x$ implies that $C \subseteq (x \cap E_v) \cup \{v\} = (x' \cap E_v) \cup \{u, v\}$. Let $v'$ be the first element after $v$ that $x$ and $x'$ differ at and assume w.l.o.g. $v' \in x \setminus x'$, we have $(x' \cap E_{v'}) \cup \{u\} = (x \cap E_{v'}) \cup \{v\}$ and since $v'$ is not added into $x'$ before Greedy terminates, there must be another circuit in $(x' \cap E_{v'}) \cup \{v'\} \subset x \cup \{v\}$ containing $v'$, which is distinct from the unique circuit $C$. The contradiction implies that $x$ and $x'$ do not differ after $v$, so $x \otimes x' = \{u, v\}$. $\qquad\square$

*Proof of Theorem 2.* We define $l_c := (1 - c)a + cb$ for $c \in [0, 1]$, let $0 \le \theta \le \theta' \le 1$ where $\pi_{l_\theta}$ and $\pi_{l_{\theta'}}$ are an adjacent swap apart[8] and the Greedy solutions on them, $x$ and $x'$, are such that $|x \otimes x'| = 2$. Let $u, v \in E$ where $x \cap \{u, v\} = \{u\}$ and $x' \cap \{u, v\} = \{v\}$, Lemma 3 implies $\pi_{l_\theta}(u) < \pi_{l_\theta}(v)$ and $\pi_{l_{\theta'}}(u) > \pi_{l_{\theta'}}(v)$, so $\pi_a(u) < \pi_a(v)$. This means as the trade-off moves from $a$ to $b$, the Greedy solution minimizing the scalarized weight changes incrementally by having exactly one element shifted to the right on $\pi_a$ (to a position not occupied by the current solution). Since at most $hm - h(h+1)/2$ such changes can be done sequentially, Greedy produces at most $hm - h(h+1)/2 + 1$ distinct solutions in total across all trade-offs between $a$ and $b$.

To show this upper bound, we keep track of the following variables as the trade-off moves from $a$ to $b$. Since each solution contains $n$ elements, let $p_i$ be the $i$th leftmost position on $\pi_a$ among those occupied by the current Greedy solution for $i = 1, \ldots, n$, we see that upon each change, there is at least a $j \in \{1, \ldots, n\}$ where $p_j$ increases. Furthermore, for all $i$, $p_i$ can increase by at most $m - n$ since it cannot be outside of $[i, m - n + i]$, so the quantity $p := \sum_{i=1}^n p_i$ can increase by at most $n(m - n)$. We see that $p$ increases by $l$ when the change is incurred by a swap in the Greedy solution such that the added element is positioned $l$ to the right of the removed element on $\pi_a$, we call this a $l$-move. Furthermore, each element pair participates in at most one swap, so $p$ can be increased by at most $m - l$ $l$-moves for every $l = 1 \ldots, m - 1$. Therefore, to upper bound the number of moves, we can assume smallest possible distance in each move, and the increase in $p$ from using all possible $l$-moves for all $l = 1, \ldots, h$ is $\sum_{j=1}^h j(m - j) \ge n(m - n)$. This means no more than $\sum_{j=1}^h (m - j) = hm - h(h+1)/2$ moves can be used to increase $p$ by at most $n(m - n)$. $\qquad\square$

---

[7]This is also done for any $\Delta_{i,j}$ containing both $a$ and $a'$.

[8]If $\theta = \theta'$, we assume $\pi_{l_\theta}$ is closer to $\pi_a$ in Kendall distance.

*Proof of Lemma 4.* First, we observe that a set supersets a base iff its rank is $n$. We see that for all $\lambda \in [0, 1]$ and $x, y \in \{0, 1\}^m$, $r(x) > r(y)$ implies $f_\lambda(x) < f_\lambda(y)$. Thus, for each $\lambda \in \Lambda$, MOEA/D performs (1+1)-EA search toward a base's superset with fitness $f_\lambda$, which concludes in $O(m \log n)$ expected steps [26]. The claim follows from the fact that MOEA/D produces $|\Lambda|$ search points in each step. $\qquad\square$

*Proof of Theorem 3.* We assume each solution in $P_\lambda$ supersets a base for all $\lambda \in \Lambda$; this occurs within expected time $O(|\Lambda| m \log n)$, according to Lemma 4. Since for each $\lambda \in \Lambda$, the best improvement in $f_\lambda$ is retained in each step, the expected number of steps MOEA/D needs to minimizes $f_\lambda$ is at most the expected time (1+1)-EA needs to minimizes $f_\lambda$. We thus fix a trade-off $\lambda$ and assume the behaviors of (1+1)-EA. Note that we use $d_\lambda \cdot w^{(\lambda)}$ in the analysis instead for integral weights; we scale $f_\lambda$ and $OPT_\lambda$ accordingly.

We call the bit flips described in Lemma 5 *good flips*. Let $s$ be the current search point, if good 1-bit flips incur larger total weight reduction than good 2-bit flips on $s$, we call $s$ 1-step, and 2-step otherwise. If at least half the steps from $s$ to the MWB $z$ are 1-steps, Lemma 5 implies the optimality gap of $s$ is multiplied by at most $1 - 1/2(m - n)$ on average after each good 1-bit flip. Therefore, from $f_\lambda(s) \leq d_\lambda(m - n)w_{max} + OPT_\lambda$, the expected difference $D_L$ after $L$ good 1-bit flips is $E[D_L] \leq d_\lambda(m - n)w_{max}(1 - 1/2(m - n))^L$. At $L = \lceil (2 \ln 2)(m - n) \log(2d_\lambda(m - n)w_{max} + 1) \rceil$, $E[D_L] \leq 1/2$ and by Markov's inequality and the fact that $D_L \geq 0$, $\Pr[D_L < 1] \geq 1/2$. Since weights are integral, $D_L < 1$ implies that $z$ is reached. The probability of making a good 1-bit flip is $\Theta((m - n)/m)$, so the expected number of steps before $L$ good 1-bit flips occur is $O(Lm/(m - n)) = O(m(\log(m - n) + \log w_{max} + \log d_\lambda))$. Since 1-steps take up most steps between $s$ and $z$, the bound holds.

If at least half the steps from $s$ to $z$ are 2-steps, Lemma 5 implies the optimality gap of $s$ is multiplied by at most $1 - 1/2n$ on average after each good 2-bit flip. Repeating the argument with $L = \lceil (2 \ln 2)n \log(2d_\lambda(m - n)w_{max} + 1) \rceil$ and the probability of making a good 2-bit flip being $\Theta(n/m^2)$, we get the bound $O(m^2(\log(m - n) + \log w_{max} + \log d_\lambda))$. Summing this over all $\lambda \in \Lambda$ gives the total bound. $\qquad\square$

*Proof of Theorem 4.* From Corollary 2, to collect a new point in $C$, it is sufficient to perform a 2-bit flip on some supported solution. In worst-case, there can be only one trade-off $\lambda \in \Lambda$ such that all non-extreme supported solutions minimize $w^{(\lambda)}$, so the correct solution is mutated with probability at least $1/l$ in each iteration, where $l$ is the number of already collected points. Since $|\Lambda|$ search points are generated in each iteration, the expected number of search points required to enumerate $C$ is $O(|\Lambda|m^2 \sum_{l=1}^{|C|} l) = O(|\Lambda||C|^2 m^2)$. $\qquad\square$

