# OpenReview forum: "Rigorous Runtime Analysis of MOEA/D for Solving Multi-Objective Minimum Weight Base Problems"
_NeurIPS.cc/2023/Conference — NeurIPS 2023 poster_

### Official Review · Reviewer_RGyY · 2023-07-01

**Soundness:** 3 good
**Presentation:** 3 good
**Contribution:** 3 good
**Rating:** 7
**Confidence:** 4

**Summary:**

This paper studies the "multi-objective minimum weight base" problem in the MOEA/D framework and proposed several useful properties.



**Strengths:**

This paper gives the running time analysis for a special kind of MOEAD problem. This is an important contribution for the MOO field since MOEAD is a very basic idea.
The strength of this paper is that, the studied algorithm is very meaningful since it take cooperation into consideration. (Line 8-9 in Algorithm 3).




**Weaknesses:**

1. This paper take multi-objective minimum weight base problems as an example. This is not general however.

**Questions:**

1. This paper mainly considers the weighted sum scalarization. It is known that this method can only recover the convex part of a PF. Will the proposed method find the whole Pareto solution?

2. When the objectives are all non-convex. It is impossible to use the supportive set to approximate the whole PS. So is there any assumption for your supportive set?

3. Line 102, a support set is extreme if ..... This statement is not clear.

4. line 250, ontrade off -> on trade off?

**Limitations:**

NO.

---

> ### Author Rebuttal · Authors · 2023-08-09
>
> We appreciate the suggestions which help us revise our work. Below, we address the questions raised:
>
> >This paper mainly considers the weighted sum scalarization. It is known that this method can only recover the convex part of a PF. Will the proposed method find the whole Pareto solution?
>
> MOEA/D can achieve this with elitism and randomization, i.e., there is always a non-zero chance of generating a non-dominated solution every time mutation operation occurs, and this solution is kept in the returned set S. However, we do not yet have sufficient insight into the underlying structures of the PF to determine how many iterations it would need.
>
> >When the objectives are all non-convex. It is impossible to use the supportive set to approximate the whole PS. So is there any assumption for your supportive set?
>
> Our Theorem 1 asserts the extreme points on the convex hull are sufficient for approximating all solutions, including the entire PF (within a factor equal the number of objectives k). This holds for all non-negative multi-objective minimization problems (i.e. no other assumption needed), and under the definition of approximation we considered. To our knowledge, this is the only published formal notion of approximation ratio in multi-objective optimization. Additive error notions like epsilon-dominance are not conducive to meaningful worst-case analyses without strong assumptions on the objectives.
>
> >Line 102, a support set is extreme if ..... This statement is not clear.
>
> Firstly, the statement implies a supported solution can be an extreme solution. Secondly, we regard operators argmin and argmax to return a set rather than an element to account for multiple minima and maxima, respectively. Finally, we use f(z)=f(x) literally: all objective values are equal. This statement excludes any solution whose image is in the interior of some facet of the convex hull, as there are always vertices on the same facet distinct from said image. Note these vertices are images of solutions minimizing the weight that is scalarized by the trade-off normal to this facet, so the condition is violated.

---

> > ### Comment · Reviewer_RGyY · 2023-08-16
> > **For Q1**
> >
> > My concern for Q1 is that, when the PF is concave, like f_1 = 1 - f_2^2.
> >
> > You can check that, using any preference w=[0,1] from w = [1,0], the minimal value of w^T f will only lead to the two end point (0,1) or (1,0).
> > Am I correct? So when the PF is concave, using LS will only find some partial solutions. Will this argument hurt your conclusion?

---

> > > ### Comment · Reviewer_RGyY · 2023-08-16
> > > **further**
> > >
> > > elitism and randomization will not help too much to find the concave part of a PF using LS. (According to my experience)

---

> > > ### Author Response · Authors · 2023-08-16
> > >
> > > This does not hurt our conclusion, as the standard bit mutation generates any solution y from any solution x with a positive probability. Furthermore, the returned set S keeps all non-dominated points found during the search, thus is not affected by the decomposition scheme such as LS, or the shape of the PF. In this example, any non-dominated point aside from (0,1) and (1,0), once reached by mutation, will be stored in S until the termination of the run. Note the earliest variants of MOEA/D already keep all non-dominated points (referring to [Zhang, et al. 2007](https://doi.org/10.1109/TEVC.2007.892759)).
> > >
> > > In our pseudo-code of the MOEA/D (line 12), S is only affected by function g, which is a vector function not reliant on weight scalarization. We do not see ambiguity in the notations here.

---

> > > > ### Comment · Reviewer_RGyY · 2023-08-16
> > > > **reply**
> > > >
> > > > I do not very agree with you. Only relying on ep seems too weak. When applying for 3-4 objective the ep will not work well.
> > > > You can check the source code of Modern software s like pymoo and plate mo. I am very familiar with these. They all  do not rely on EP.
> > > > My understanding is that when using ep the alg works like random search. You can check some recent works from Qian Chao from NIU. Their findings is mainly like if the store ability is strong the the alg can find the global opt.
> > > > But I think relying EP analyzing MOEAD is not good. EP is not the essential part of why MOEAD work. Very many new works avoid using it.

---

> > > > > ### Author Response · Authors · 2023-08-20
> > > > >
> > > > > Thanks for the additional comment. We have proven that computing the EPs leads to a good approximation for the problems we considered. We agree that this approach may be further enhanced with additional operators but proving that this also leads to a better approximation behavior may be extremely difficult and can be considered as a topic for future work.

---

> > > > > > ### Comment · Reviewer_RGyY · 2023-08-20
> > > > > > **reply**
> > > > > >
> > > > > > Thanks for your clarification. I am pleased and hold positive view as the current version. I keep my score.
> > > > > > I encourage the authors try different scalarization function in further works.
> > > > > > I have no additional comments.

---

### Official Review · Reviewer_EzWR · 2023-07-04

**Soundness:** 3 good
**Presentation:** 2 fair
**Contribution:** 3 good
**Rating:** 6
**Confidence:** 4

**Summary:**

This paper presents a comprehensive theoretical and empirical analysis of the Multi-Objective Evolutionary Algorithm based on Decomposition (MOEA/D) for the Multi-Objective Minimum Weight Base problem. The authors provide a detailed explanation of the algorithm, including its fitness functions and decomposition scheme. They also propose a series of algorithms for finding extreme points and a complete trade-off set. The paper includes a rigorous theoretical analysis, providing expected time bounds for the algorithm to minimize scalarized weights and enumerate the convex hull of the feasible region. The authors also conduct an experimental investigation on various bi-objective minimum spanning tree instances, comparing the performance of MOEA/D against GSEMO. The results demonstrate that MOEA/D significantly outperforms GSEMO in computing extreme points under an appropriate decomposition.

**Strengths:**

1.	The paper provides a thorough theoretical analysis of the MOEA/D algorithm, including expected time bounds for key operations, which contributes to the understanding of the algorithm's performance and efficiency.
2.	The authors apply their theoretical findings to a real-world problem demonstrating the practical relevance of their research.
3.	The paper includes a comparative analysis of MOEA/D and GSEMO, and provides valuable insights into the strengths and weaknesses of these algorithms.
4.	The authors provide a detailed explanation that helps understand the MOEA/D algorithm.


**Weaknesses:**

1.	The paper makes several assumptions, such as distinct supported solutions having distinct images under the weight function, which might not be applicable in all scenarios. How would the theory guides the application of MOEA/D?
2.	The paper does not provide a detailed discussion on how to choose the parameters for the MOEA/D algorithm.
3.     This paper may only be interested by the evolutionary computation community.


**Questions:**

1.     How would the theory guides the application of MOEA/D?
2.	Could the authors elaborate on the rationale behind the assumption that distinct supported solutions have distinct images under the weight function.? Are there any scenarios where this assumption might not hold, and if so, how would this affect the results?
3.	Could the authors provide more insights into how these parameters should be selected?


**Limitations:**

Societal impact of the work is not discussed in this paper.

---

> ### Author Rebuttal · Authors · 2023-08-09
>
> We would like to address the in-depth questions here, and try to work the dicussions into our revision to improve the paper:
>
> >How would the theory guides the application of MOEA/D?
>
> Our analysis of the multi-objective min weight base problem gives guidance on how to choose scalarization weights for MOEA/D for the problem. We have demonstrated this to some degree in the experiments.
>
> >Could the authors elaborate on the rationale behind the assumption that distinct supported solutions have distinct images under the weight function.? Are there any scenarios where this assumption might not hold, and if so, how would this affect the results?
>
> Duplicate images among points on the convex hull only occur if there are at least two ground set elements having the same weight across all weight functions, i.e., at least two columns in the weight matrix w that are identical (this is a consequence of the fact that Greedy can compute all min weight bases). We elaborated in Section 3 that the existence of such elements does not affect the results since these are not concerned with enumerating all solutions, only those whose images are distinct, i.e., enumerating points in the objective space. To be specific, such elements might spawn additional min weight bases for some scalarized weight, whose pairwise xor are contained within these elements (thus sharing images). By fixing the relative ordering among these elements on the sorting permutations, such differences effectively vanish, i.e., these additional bases disappear. Since fixing these orderings is always valid for all scalarized weights, we can assume such elements do not exist.
>
> We further note that there can be multiple such "classes" of elements (elements in the same class share weights while those in different classes do not). The presence of multiple such classes does not affect things either, because swapping the relative ordering between two classes on a permutation can always be done via a sequence of adjacent swaps, all while preserving elements' ordering within each class at every step (thus fixing orderings as mentioned). This process can be generalized to more than two classes using only adjacent swaps, e.g., by swapping one pair of classes at a time. All this means, again, that we can w.l.o.g. fix the orderings among same-weight elements.
>
> Regarding the additional min weight bases we mentioned, the permutations that lead to Greedy producing these bases only differ in the orderings among same-weight elements. As we mentioned in Section 3, we can always transform one into the other via a adjacent swap sequence such that every intermediate step returns a sorting permutation. As such, even in the presence of such elements, the set of convex hull solutions is still 2-Hamming connected as a consequence of our Lemma 3 (hence Corollary 2).
>
> >Could the authors provide more insights into how these parameters should be selected?
>
> We can only comment on the task of finding points on the convex hull of the PF. For the scalarization trade-offs, we note that it is optimal to have exactly one trade-off per extreme point (if we only care about extreme points), as this would not incur redundant function evaluations (aside from those arising from randomization). We included a discussion on how to find these trade-offs in Section 5, e.g., processing the output of the schemes in Section 4. If we want to find all convex hull points rather than just extreme points, we need not process this output, and each trade-off would be normal to a distinct facet of the convex hull.
>
> As for the neighborhood size, we remark that greater collaborations among solutions help reduce the number of iterations needed, at the cost of making each iteration more costly, e.g., from more comparisons. We do not have a meaningful theoretical result on the impact of this parameter on MOEA/D's runtime.

---

> > ### Comment · Reviewer_EzWR · 2023-08-16
> > **Thanks for the response**
> >
> > Thank all the authors for their detailed responses. I think the anwsers are reasonable.

---

### Official Review · Reviewer_vbRF · 2023-07-06

**Soundness:** 3 good
**Presentation:** 2 fair
**Contribution:** 3 good
**Rating:** 7
**Confidence:** 3

**Summary:**

The result is for the cross product of a problem class and algorithm. The problem is the multi-objective weight base problem and the algorithm is Multi-objective, evolutionary algorithm with decomposition (MOEA/D). This algorithm effectively optimizes by decomposing the objectives into single-objectives. With an appropriate decomposition, the algorithm can find all extreme points with expected fixed-parameter polynomial time.  The problem has the appropriate decomposition. Experiments agree with the theory and show the algorithm is faster than another EA called GSEMO

**Strengths:**

The problem is in the matroid optimization category and generalizes wrt a previously studied bi-objective minimum spanning tree problem.

Decomposition into single-objective problems is one of two well-used approaches to Multi-objective optimization (MOO) so the MOEA/D is relevant.

Prior state of art is for a different (simplified) algorithm (GSEMO) which shows a pseudo-polynomial run-time for 2 not k objectives.
Together with recent analyses of NSGA-II, the paper represents significant progress in developing runtime analysis of practically relevant multi-objective evolutionary algorithms.

The paper develops insight into the local structure (Hamming neighborhoods) of matroid optimization problems. This structural
insights will be helpful to analyse the runtime of other evolutionary algorithms on such problems.

This is the first rigorous runtime analysis of MOEA/D. The authors provide upper bounds on the expected
time to enumerate the convex hull of the Pareto front of the bi- and k-objective minimum weight base (MWB) problem. The assumption is that the algorithm is provided with optima for complete trade-off sets (e.g.,via the Greedy algorithm).

**Weaknesses:**

One problem with the paper is readability. Algorithm and problem names  are stated early as are comparisons but they are poorly described until the preliminaries.  There are a lack of clear intuitive descriptions of the problem, the algorithm (and those it is compared with) within the preliminaries. The reader has to jump across statements to figure out the analytic approach and to understand how/why it works. Granted it's challenging to describe a matroidal problem and one that is multi-objective, but the writing does not offer sufficient help with this.  The problem definition is inline on line 83 for example.

The objectives arise from the problem's matroidal properties and in a sense, the problem formulation makes it easy for each objective to be solved for extrema.  The challenge with MOEA/D in practice is the decomposition and this is neither noted nor related to the matroidal problem analyzed here.  Are there real world problems that are as ideally decomposable?  The analysis depends on a property of the problem (linear independence), the convex hull of its non-dominated front, and the ability to compute extrema for more than two objectives.
It is not explained why the MOEA/D is a good algorithm to use vs a dominance-based one.
The stochastic behavior of the MOEA/D algorithm has not been accounted for.
It would be helpful to additionally provide an analysis for standard multi-objective benchmark functions to allow for a comparison with analyses of other MOEAs.
The description of the experimental setting lacks detail.












**Questions:**

Why are there no figures!?
How does this problem --  the multi-objective weight base problem relate to common MOO problems? What are the properties of a harder problem that we would use the algorithm for?
How is the first generation population of GSEMO initialized, and does this give the MOEA/D an unfair advantage?

---

> ### Author Rebuttal · Authors · 2023-08-09
>
> We appreciate the broader perspective on our work and would like to address the concerns from a high-level view:
>
> >How does the multi-objective weight base problem relate to common MOO problems?
>
> The problem is a generalization of the well-known classical minimum weight base problem. We believe this alone makes the problem worth studying.
>
> >What are the properties of a harder problem that we would use the algorithm for?
>
> On a basic level, the algorithm is suitable for problems whose objectives are decomposable. Furthermore, the decomposition scheme based on linear scalarization is effective on any problem where the non-dominated front is close to being convex. More generally, however, wider application of MOEA/D is beyond the scope of this work, and we would refer to other works on MOEA/D.
>
> Note that we do not claim MOEA/D is the best algorithm for this problem or any related problem. Here, we study the base variant of MOEA/D, which is not tailored to any specific problem. On the other hand, establishing connections between the multi-objective weight base problem and harder problems is an interesting topic in its own right.
>
> >How is the first generation population of GSEMO initialized, and does this give the MOEA/D an unfair advantage?
>
> GSEMO starts with a single solution drawn uniformly at random from the set of all bit-strings. We do not see any unfair advantage from this, as MOEA/D also starts with uniformly random bit-string solutions.

---

> > ### Comment · Reviewer_vbRF · 2023-08-17
> >
> > Thank you for your responses.  My opinion of the paper remains the same. I still feel the paper needs figures.

---

### Official Review · Reviewer_rEPr · 2023-07-13

**Soundness:** 2 fair
**Presentation:** 2 fair
**Contribution:** 2 fair
**Rating:** 4
**Confidence:** 4

**Summary:**

This paper presents the first run-time analysis of the MOEA/D for the multi-objective minimum weight base problem, e.g., multi-objective minimum spanning tree (MST). The theoretical results show the MOEA/D can find all extreme points within expected fixed-parameter polynomial time given an appropriate decomposition setting. Experiments on minimum spanning tree instances agree with the theoretical results.

**Strengths:**

1/ This paper gives the first analysis of the MOEA/D algorithm for the multi-objective minimum weight base problem.


**Weaknesses:**

1.  **Lack of novelty and originality**: Although the paper presents the first analysis of the MOEA/D algorithm for the multi-objective minimum weight base problem, it does not offer significant novelty in the context of multi-objective combinatorial optimization. Recent studies, such as the analysis of NSGA-II for similar problems, have already explored this area.

2. **Insufficient innovation in analysis techniques**: The technical approach used in this paper is similar to previous work, focusing on enumerating the convex hull Conv($𝐹$) of the non-dominated front. The paper does not introduce substantial innovations or improvements to the existing techniques, which limits its contribution to the field.

3. **Limited insights from the results**: The results obtained in the paper do not provide new insights or implications for algorithm design. Although the MOEA/D algorithm is shown to be faster than the GSEMO algorithm for the given problem, this alone is not enough to warrant a significant contribution to the research community.

**Questions:**

Why do you claim “multi-objective minimum weight base problem is a significant generalization of the previously studied bi-objective minimum spanning tree problem”? Please give more disccusions.

**Limitations:**

No, this paper does not include the limitations.

---

> ### Author Rebuttal · Authors · 2023-08-09
>
> We believe we have included the answer to this question in Section 2 of our paper as it was submitted. Nevertheless, we will address it here, and try to make the paper clearer to unfamiliar readers:
>
> >Why do you claim “multi-objective minimum weight base problem is a significant generalization of the previously studied bi-objective minimum spanning tree problem”?
>
> Spanning forests in an undirected graph are equivalent to bases in a graphic matroid, whose indepedent sets correspond to edge sets of acyclic subgraphs in said graph. This fact, which we mentioned in Section 2 citing examples of matroid types, has long been known in combinatorics. Note that we cited past runtime analyses on matroid optimization in our paper (e.g. Reichel et. al., 2008), so this has also been known within the runtime analyses literature.
>
> On the other hand, "multi-objective" generalizes "bi-objective". To our knowledge, all previous theoretical runtime analyses (including ones for NSGA-II) did not consider more than two objectives, so our findings are the first attempt at a formal generalization in this direction.
>
> For these reasons, the problem we study subsumes bi-objective MST, the latter of which represents a tiny fraction of the former. Consequently, our results also hold for matroids not equivalent to any graphic matroid, thus strictly generalize runtime analyses on spanning trees.
>
> >The paper does not introduce substantial innovations or improvements to the existing techniques.
>
> We analyzed the convex hull of the non-dominated front via looking at actions on permutations over the ground set, bypassing the objective space altogether and allowing easy generalization to more objectives, while leading to smaller upper bounds (e.g. Theorem 2). We have not seen existing studies on the problem (or any of its special cases for that matter) employing similar techniques.

---

> > ### Comment · Reviewer_rEPr · 2023-08-16
> >
> > My opinion of the paper has not changed in light of the authors' response. I will keep my current score for this paper.

---

### Official Review · Reviewer_8vmn · 2023-07-17

**Soundness:** 3 good
**Presentation:** 2 fair
**Contribution:** 3 good
**Rating:** 6
**Confidence:** 3

**Summary:**

This paper provides a run-time analysis of the MOEA/D algorithm for the multi-objective minimum weight base problem. It focuses on approximating the non-dominated front and shows that MOEA/D obtains a factor 2-approximation for two objectives in expected polynomial time. Empirical studies are conducted to verify the theoretical results.

**Strengths:**

1. A run-time analysis is presented on the MOEA/D algorithm, a classical multi-objective evolutionary algorithm that has rarely been analyzed theoretically.
2. It is shown that MOEA/D obtains a factor 2-approximation for two objectives in expected polynomial time, compared with a pseudo-polynomial run-time of an existing method.

**Weaknesses:**

1. The definition of the multi-objective minimum weight base problem is unclear, e.g., the range of the decision variables and what the objectives are.
2. The empirical study is performed by comparing with GSEMO, an algorithm published in 2007. This might be inefficient in verifying the superiority of the analyzed MOEA/D.

**Questions:**

1. In the experiments, why not consider more state-of-the-art algorithms on bi-objective minimum spanning tree instances?
2. According to the definition of dominance relation in Section 2, solutions with objectives taking the same value will dominate each other, which can cause a contradiction. How is this situation handled in the algorithm?
It says that “Such a set is known to guarantee a 2-approximation of the non-dominated set for k= 2”. Nevertheless, this conclusion was obtained on the multi-objective minimum spanning tree problem; does it still hold on to the more general multi-objective minimum weight base problem?

**Limitations:**

It is suggested to analyze the front shape of the tested problems, e.g., if problems with irregular Pareto fronts have been examined.

---

> ### Author Rebuttal · Authors · 2023-08-09
>
> >In the experiments, why not consider more state-of-the-art algorithms on bi-objective minimum spanning tree instances?
>
> GSEMO is a state of the art algorithm in the theoretical analysis of evolutionary multi-objective algorithms and a wide range of rigorous runtime results have been obtained for GSEMO. For GSEMO, previous theoretical runtime results showing a pseudo-polynomial time to compute a 2-approximation have been obtained. Other popular evolutionary multi-objective algorithms such as NSGA-II, IBEA, or SPEAS2 do not provide such good approximation guarantees, or have even higher computational cost than GSEMO to achieve a 2-approximation.
> Therefore, we think that the comparison to GSEMO is valid and important and clearly show the strength of the MOEA/D approach.
>
> >According to the definition of dominance relation in Section 2, solutions with objectives taking the same value will dominate each other, which can cause a contradiction. How is this situation handled in the algorithm?
>
> We use the classical definition of dominance in the multi-objective optimization literature. In MOEA/D, duplicate objective values are handled at line 11: the algorithm arbitrarily chooses one solution to keep, discarding the rest.
>
> >It says that “Such a set is known to guarantee a 2-approximation of the non-dominated set for k= 2”. Nevertheless, this conclusion was obtained on the multi-objective minimum spanning tree problem; does it still hold on to the more general multi-objective minimum weight base problem?
>
> Yes. We proved in Theorem 1 that it gives a k-approximation for any k-objective instance. This applies to all multi-objective minimization problems, not just multi-objective minimum weight base. Note the cited 2-approximation result here also applies to all bi-objective problems, despite the original author's claim being restricted to spanning trees.

---

### Author Rebuttal · Authors · 2023-08-02

We thank the reviewers for their reviews and comments. We address the specific comments and questions raised by each reviewer in individual responses.

We see some confusion regarding approximating the non-dominated front, specifically our Theorem 1. We have prefaced the theorem by stating that the result is not restricted to multi-objective minimum weight base, and we have minimized the assumptions used in the theorem itself to emphasize this. Nevertheless, we will try to clarify it further in our revision.

---

### Decision · Program_Chairs · 2023-09-21

**Decision:**

Accept (poster)

**Comment:**

The paper provides a thorough theoretical analysis of MOEA/D and applies it to Multi-Objective Minimum Weight Base problem. The reviewers generally agree that the paper is technically solid, though some have raised concerns about the assumptions made in the paper and the limitations of the approach. The authors have provided rebuttals to address most of the concerns, meanwhile there are still concerns that are not fully addressed, including the similarity between the studied problem and the previously studied minimum spanning tree problem.